# Urate-lowering therapy may mitigate the risks of hospitalized stroke and mortality in patients with gout

Fu-Shun Yen[1], Chih-Cheng Hsu[2,3,4], Hsin-Lun Li[5,6], James Cheng-Chung Wei[7,8,9]*, Chii-Min Hwu [10,11]*

1 Dr. Yen's Clinic, Taoyuan City, Taiwan, 2 Institute of Population Health Sciences, National Health Research Institutes, Zhunan, Miaoli County, Taiwan, 3 Department of Health Services Administration, China Medical University, Taichung, Taiwan, 4 Department of Family Medicine, Min-Sheng General Hospital, Taoyuan, Taiwan, 5 Management Office for Health Data, China Medical University, Taichung, Taiwan, 6 Graduate Institute of Clinical Medical Science, College of Medicine, China Medical University, Taichung, Taiwan, 7 Institute of Medicine, Chung Shan Medical University, Taichung City, Taiwan, 8 Department of Medicine, Chung Shan Medical University Hospital, Taichung City, Taiwan, 9 Graduate Institute of Integrated Medicine, China Medical University, Taichung, Taiwan, 10 Faculty of Medicine, National Yang-Ming University School of Medicine, Taipei, Taiwan, 11 Section of Endocrinology and Metabolism, Department of Medicine, Taipei Veterans General Hospital, Taipei, Taiwan

* wei3228@gmail.com (JCCW); chhwu@vghtpe.gov.tw (CMH)

## Abstract

### Objectives

Although studies have demonstrated the association of hyperuricemia with cardiovascular (CV) diseases, few have explored the effect of urate-lowering therapy (ULT) on the incidence of CV diseases. Therefore, we compared the risks of hospitalized coronary artery disease (CAD), stroke, heart failure (HF), and all-cause mortality between ULT users and nonusers among patients with gout.

### Methods

We performed this retrospective cohort study using Taiwan's population-based National Health Insurance Research Database. In total, 5218 patients with gout were included from 2000 to 2012. We compared the incidence rates (IRs) of hospitalized CAD, stroke, HF, and all-cause mortality between ULT users and matched nonusers.

### Results

The IRs of hospitalized stroke were 0.6 and 1.0 per 100 person-years for ULT users and nonusers, respectively, after adjusting for age, sex, residence, comorbidities, and medications. ULT users showed lower adjusted hazard ratios (aHR) for hospitalized stroke (aHR: 0.52, $p < 0.001$) and all-cause mortality (aHR: 0.6, $p = 0.02$) than nonusers. Subgroup analyses revealed that uricosuric agents and xanthine oxidase inhibitors were significantly associated with lower risks of hospitalized stroke and all-cause mortality, respectively. The effect of uricosuric agents on the decrease in hospitalized stroke risk indicated a dose–response relationship.

**Data Availability Statement:** Data are available from the National Health Insurance Research Database (NHIRD) published by Taiwan National Health Insurance (NHI) Bureau. The data utilized in this study cannot be made available in the paper,

the supplemental files, or in a public repository due to the "Personal Information Protection Act" executed by Taiwan's government, starting from 2012. Requests for data can be sent as a formal proposal to the NHIRD (http://nhird.nhri.org.tw) or by email to nhird@nhri.org.tw.

**Funding:** This work was supported in part by the Taiwan Ministry of Health and Welfare Clinical Trial Center (MOHW108-TDU-B-212-133004), China Medical University Hospital, Academia Sinica Stroke Biosignature Project (BM10701010021), MOST Clinical Trial Consortium for Stroke (MOST 107-2321-B-039 -004-), Tseng-Lien Lin Foundation (Taichung, Taiwan), and Katsuzo and Kiyo Aoshima Memorial Funds (Japan), The funders had no role in study design, data collection and analysis, decision to publish, or preparation of the manuscript.

**Competing interests:** The authors have declared that no competing interests exist.

## Conclusions

Our study showed lower risks of hospitalized stroke and all-cause mortality in ULT users than in nonusers among patients with gout. Therefore, patients with gout may receive ULT to mitigate the risks of hospitalized stroke and mortality.

## Introduction

Serum uric acid (SUA) has been reported to be related closely to cardiovascular (CV) diseases since the nineteenth century [1]. After years of research and relevant advances, the relationship between SUA and CV diseases has been elucidated recently. A meta-analysis reported increased risk ratios of 1.22 and 1.12 for coronary artery disease (CAD) in women and men with hyperuricemia, respectively [2]. Another meta-analysis of prospective cohort studies revealed a 12% increase in mortality with every 1 mg/dL increase in uric acid (UA) among patients with CAD [3]. Hyperuricemia is also associated with a high risk of incident heart failure (HF), which follows a dose–response trend [4]. Few studies have investigated the relationship between SUA and stroke. Only Kim et al. conducted a systemic review establishing that hyperuricemia induced higher risks of stroke incidence (relative risk, 1.41) and mortality (relative risk, 1.26) [5]. Conversely, some studies have reported that because hyperuricemia always appeared with several metabolic diseases, after controlling for these confounders, no significant association was observed between hyperuricemia and CV diseases [2, 6, 7]. Therefore, hypouricemic intervention studies should be conducted to clarify whether the high level of SUA is the cause of or an accompanying condition in atherosclerotic CV diseases.

Some animal studies and randomized clinical trials in adolescents with early hypertension (HT) have shown that urate-lowering therapy (ULT) significantly decreased blood pressure [8, 9]. HT is one of the most common causes of stroke, and stroke is a leading cause of mortality [10], with more than half of stroke survivors experiencing severe and permanent disability [11]. By decreasing blood pressure, ULT is a potential promising strategy to prevent the development or progression of stroke. Few studies have reported the benefits of allopurinol use in preventing CV events and increasing survival [12–14]. Because the effect of ULT on the risks of CV diseases and mortality remains inconclusive, we conducted this retrospective cohort study to evaluate the risks of hospitalized stroke, CAD, HF, and all-cause mortality between ULT users and nonusers in patients with gout.

## Materials and methods

### Data source

In 1995, the Taiwanese government initiated the single-payer National Health Insurance (NHI) program to increase healthcare accessibility, and physicians are required to provide the claims data of each patient visit to the NHI Administration. Thus, the NHI Research Database (NHIRD) contains the medical utilization records of insured persons as well as their information on location, sex, age, investigations, diagnoses, prescriptions, and details of each outpatient or inpatient visit [15]. The International Classification of Diseases, Ninth Revision, Clinical Modification (ICD-9-CM) was used to code disease diagnoses. The National Health Research Institutes has made available to researchers a randomly sampled dataset from 2000 to 2013 including the data of 1 million individuals (LHID 2000), which was used in this study. This study was approved by the Research Ethics Committee of China Medical University and

Hospital (CMUH-104-REC2-115). Because the identifying information in the NHIRD was de-identified and encrypted, we were exempt from obtaining informed consent.

## Study design

From the NHIRD 2000, we selected patients with gouty arthritis (ICD-9-CM: 274.00–274.03 and 274.8–274.9) aged between 20 and 79 years from 2000 to 2012 as our study population. Patients who had received ULT with allopurinol (M04AA01), febuxostat (M04AA03), benz-bromarone (M04AB03), probenecid (M04AB01), and sulfinpyrazone (M04AB02) were included in the case cohort, whereas the control cohort included patients who were diagnosed as having gouty arthritis and were not prescribed ULT during the follow-up period. The main outcomes were all-cause mortality, hospitalized CAD, hospitalized stroke, and hospitalized HF. We used the date of first ULT and a date randomly assigned within the observation period as the index dates for the case and control cohorts, respectively. The exclusion criteria were patients aged less than 20 years or more than 80 years; who died or were hospitalized with the diagnosis of CAD, stroke, or HF within 6 months after the index date; who underwent chemotherapy or were diagnosed as having cancer by the index date; and who had missing data on sex, age, or region. The study population was followed up until patients were admitted for CAD, stroke, or HF; patient data were censored for death; patient withdrawal from the NHI program; or the end of 2013.

## Statistical analysis

We performed propensity score matching for age, sex, residential area, index year, and year of gouty arthritis diagnosis between cases and controls at a 1:1 ratio in a logistic regression model [16]. The standardized mean difference (SMD) was applied to the strata of sex, age, area, comorbidity, drugs, and follow-up period to verify the comparability between these two groups. We matched the frequency distribution of the case and control cohorts by sex, age, area, comorbidity [i.e., HT (ICD-9-CM: 401–405 and A26), diabetes mellitus (DM) (250.x0, 250.x2, and A181), CAD (410–414), stroke (362.34 and 430.x–438.x), HF (428), hypercholesterolemia (272, 278, and A189), peripheral vascular diseases (093.0, 437.3, 440.x, 441.x, 443.1–443.9, 47.1, 557.1, 557.9, and V43.4), atrial fibrillation (427.3), rheumatologic diseases (446.5, 710.0–710.4, 714.0–714.2, 714.8, and 725.x), renal diseases (403.01, 403.11, 403.91, 404.02, 404.03, 404.12, 404.13, 404.92, 404.93, V42.0, V45.1, V56.x, and 790), and alcohol-related diseases (291.x, 303.x, 305.0, 357.5, 425.5, 535.3, 571.0–571.3, 980.0, and E947.3)], and drugs [i.e., angiotensin-converting enzyme (ACE) inhibitors, angiotensin II receptor blockers (ARBs), β-blockers, calcium channel blockers, diuretics, potassium-sparing diuretics, other antihypertensives, metformin, sulfonylureas, insulin, statin, and aspirin]. The incidence rate (IR) of events was defined as the number of events divided by the observed person-years. To assess the dose–response trend, we analyzed the risk of hospitalized stroke according to three equally distributed cumulative durations of uricosuric agent treatment ($\leq$1, 1–5, or >5 months); and the cumulative mean defined daily dose (DDD) of uricosuric agents ($\leq$0.5, 0.5–0.8, and >0.8 DDD/month) relative to nonuse of ULT. We obtained the cumulative mean DDD by dividing the cumulative DDD by the duration of uricosuric agent use. DDD is a technical unit of measurement, defined as the assumed average daily maintenance dose for a drug. The WHO set the DDD at 100 mg, 1 g, 300 mg, 400 mg, and 80 mg for benzbromarone, probenecid, sulfinpyrazone, allopurinol, and febuxostat, respectively. We derived crude hazard ratios (cHRs), adjusted hazard ratios (aHRs), and 95% confidence intervals (CIs) using multivariable Cox proportional hazards regression models. Patient data were censored after a single defined CV event. Separate Cox models were used to evaluate the effects of ULT for different outcomes.

The Kaplan–Meier method was used to derive the cumulative incidence of hospitalized stroke and all-cause mortality in ULT users and nonusers. Their statistical significance was detected using the log-rank test. All analyses were performed using the SAS 9.4 software program (SAS Institute, Cary, NC, USA). The null hypothesis of no effective difference between the two groups was rejected if $p < 0.05$.

## Results

The flowchart for the selection of the ULT case cohort and non-ULT control cohorts from the NHIRD is shown in Fig 1. In total, we identified 2609 patients. The distribution of ULT users and nonusers before and after matching is presented in Table 1. Before matching, ULT nonusers had a lower ratio of men, were older, and had more comorbidities and shorter follow-up time than ULT users. After balancing these variables, 80.5% of the matched patients were men, and the mean age [standard deviation (SD)] of ULT users and nonusers was 48 (15.1) and 48.2 (14.9) years, respectively. The mean (SD) follow-up time of ULT users and nonusers was 4.52 (2.81) and 4.27 (2.8) years, respectively.

Table 2 shows that ULT users were 0.52 times less susceptible to the risk of hospitalized stroke compared with ULT nonusers (IR per 100 person-years: 0.6 vs 1.0, 95% CI = 0.39–0.7, $p < 0.001$). Risks of ischemic stroke (aHR: 0.41) and all-cause mortality (aHR: 0.6) were also

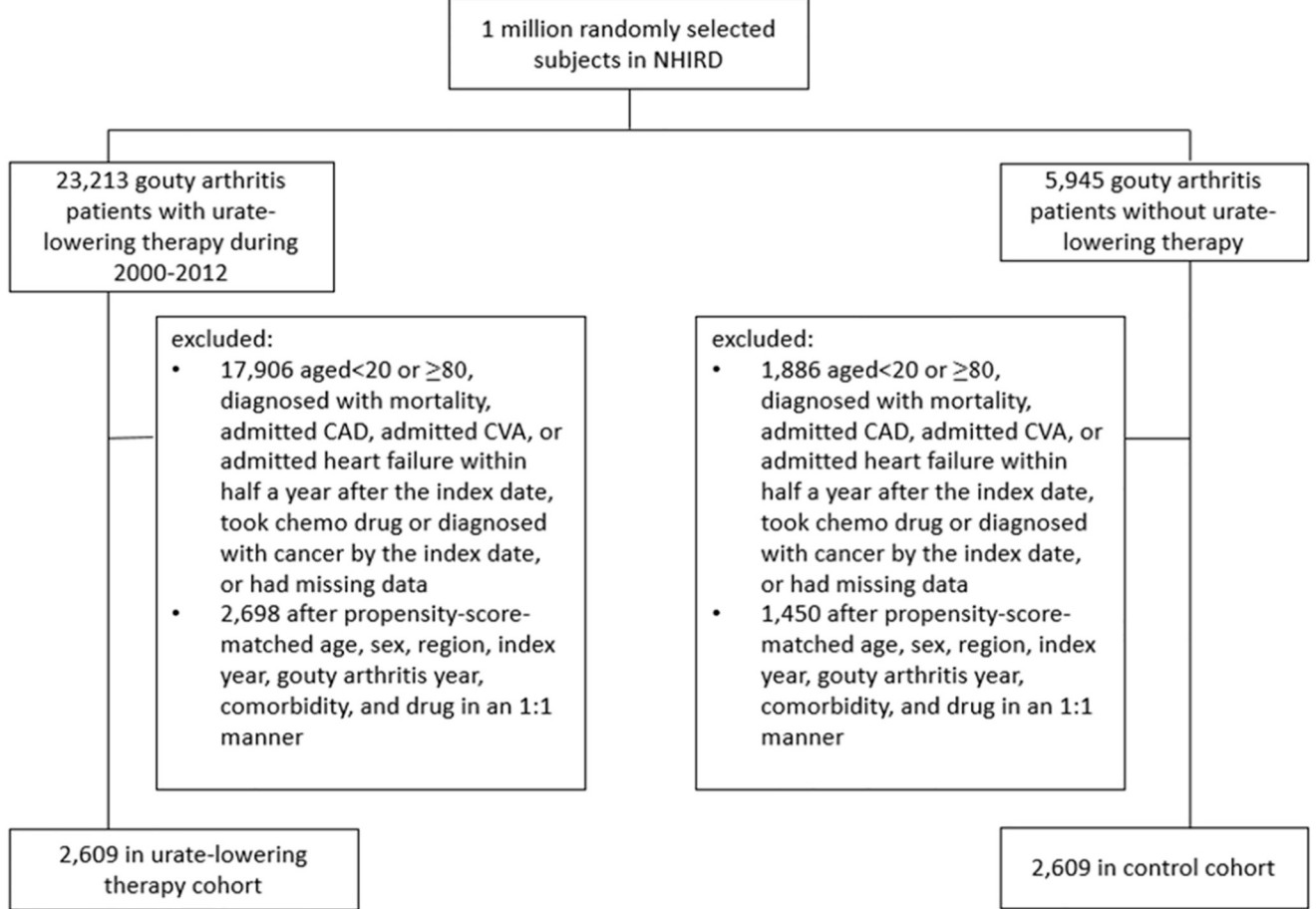

**Fig 1. Flowchart of selecting the urate-lowering therapy cohort and the control cohort from the National Health Insurance Research Database.**

**Table 1.  Baseline characteristics of the study cohorts receiving or not receiving urate-lowering therapy.**

|  | Before matching | | | | | After matching | | | | |
|  | Urate-lowering therapy | | | | | Urate-lowering therapy | | | | |
|  | No | | Yes | | | No | | Yes | | |
|  | 4059 | | 5307 | | | 2609 | | 2609 | | |
|  | n | % | n | % | SMD | n | % | n | % | SMD |
| **Sex** | | | | | | | | | | |
| Female | 1173 | 28.9 | 687 | 12.9 | 0.41 | 516 | 19.8 | 502 | 19.2 | 0.01 |
| Male | 2886 | 71.1 | 4620 | 87.1 | 0.41 | 2093 | 80.2 | 2107 | 80.8 | 0.01 |
| **Age, years** | | | | | | | | | | |
| 20–39 | 1204 | 29.7 | 2114 | 39.8 | 0.21 | 874 | 33.5 | 907 | 34.8 | 0.03 |
| 40–59 | 1804 | 44.4 | 2095 | 39.5 | 0.11 | 1131 | 43.3 | 1104 | 42.3 | 0.02 |
| 60–79 | 1051 | 25.9 | 1098 | 20.7 | 0.12 | 604 | 23.2 | 598 | 22.9 | 0.01 |
| Mean (SD) | 49.6 | 14.8 | 46.3 | 15.2 | 0.22 | 48.2 | 14.9 | 48 | 15.1 | 0.01 |
| **Area** | | | | | | | | | | |
| North | 1887 | 46.5 | 2447 | 46.1 | 0.01 | 1189 | 45.6 | 1179 | 45.2 | 0.01 |
| Central | 865 | 21.3 | 1158 | 21.8 | 0.01 | 556 | 21.3 | 548 | 21 | 0.01 |
| South | 1180 | 29.1 | 1497 | 28.2 | 0.02 | 771 | 29.6 | 793 | 30.4 | 0.02 |
| Other | 127 | 3.1 | 205 | 3.9 | 0.04 | 93 | 3.6 | 89 | 3.4 | 0.01 |
| **Comorbidity** | | | | | | | | | | |
| Hypertension | 1213 | 29.9 | 1648 | 31.1 | 0.03 | 809 | 31 | 801 | 30.7 | 0.01 |
| DM | 459 | 11.3 | 394 | 7.4 | 0.14 | 239 | 9.2 | 248 | 9.5 | 0.01 |
| CAD | 491 | 12.1 | 515 | 9.7 | 0.08 | 293 | 11.2 | 293 | 11.2 | <0.001 |
| Stroke | 273 | 6.7 | 305 | 5.7 | 0.04 | 176 | 6.7 | 176 | 6.7 | <0.001 |
| Heart failure | 100 | 2.5 | 115 | 2.2 | 0.02 | 61 | 2.3 | 65 | 2.5 | 0.01 |
| Hypercholesterolemia | 1165 | 28.7 | 1390 | 26.2 | 0.06 | 709 | 27.2 | 702 | 26.9 | 0.01 |
| Peripheral vascular diseases | 135 | 3.3 | 109 | 2.1 | 0.08 | 64 | 2.5 | 66 | 2.5 | 0.005 |
| Atrial fibrillation | 35 | 0.9 | 43 | 0.8 | 0.01 | 24 | 0.9 | 25 | 1 | 0.004 |
| Rheumatologic diseases | 172 | 4.2 | 99 | 1.9 | 0.14 | 76 | 2.9 | 66 | 2.5 | 0.02 |
| Renal diseases | 121 | 3 | 182 | 3.4 | 0.03 | 88 | 3.4 | 87 | 3.3 | 0.002 |
| Alcohol-related diseases | 131 | 3.2 | 170 | 3.2 | 0.001 | 84 | 3.2 | 94 | 3.6 | 0.02 |
| CCI(SD) | 0.27 | 0.77 | 0.20 | 0.67 | 0.11 | 0.24 | 0.73 | 0.25 | 0.76 | 0.01 |
| **Drug** | | | | | | | | | | |
| ACE inhibitors/ARBs | 774 | 19.1 | 1054 | 19.9 | 0.02 | 514 | 19.7 | 519 | 19.9 | 0.005 |
| β-blockers | 1325 | 32.6 | 1553 | 29.3 | 0.07 | 822 | 31.5 | 813 | 31.2 | 0.01 |
| Calcium-channel blockers | 1069 | 26.3 | 1385 | 26.1 | 0.01 | 702 | 26.9 | 702 | 26.9 | <0.001 |
| Diuretics | 940 | 23.2 | 1108 | 20.9 | 0.06 | 584 | 22.4 | 580 | 22.2 | 0.004 |
| Potassium sparing diuretics | 120 | 3 | 98 | 1.8 | 0.07 | 64 | 2.5 | 66 | 2.5 | 0.005 |
| Other antihypertensive | 498 | 12.3 | 668 | 12.6 | 0.01 | 336 | 12.9 | 349 | 13.4 | 0.01 |
| Metformin | 275 | 6.8 | 263 | 5 | 0.08 | 160 | 6.1 | 163 | 6.2 | 0.005 |
| sulfonylurea | 284 | 7 | 297 | 5.6 | 0.06 | 171 | 6.6 | 178 | 6.8 | 0.01 |
| Insulin | 92 | 2.3 | 129 | 2.4 | 0.01 | 67 | 2.6 | 73 | 2.8 | 0.01 |
| Statin | 383 | 9.4 | 320 | 6 | 0.13 | 208 | 8 | 213 | 8.2 | 0.01 |
| Aspirin | 680 | 16.8 | 779 | 14.7 | 0.06 | 432 | 16.6 | 423 | 16.2 | 0.01 |
| mean of follow-up period of outcome | 3.58 | 2.58 | 6.66 | 3.66 | 0.95 | 4.27 | 2.8 | 4.52 | 2.81 | 0.09 |

SMD, standardized mean difference, ≤0.10 indicates a negligible difference between the two cohorts; outcomes consisting of all-cause mortality, hospitalized coronary artery disease, hospitalized stroke, and hospitalized heart failure.

Table 2. Incident rates of mortality, CAD, stroke, and heart failure between urate-lowering drug users vs. nonusers.

| Outcome | Urate-lowering therapy | | | | | | cHR (95%CI) | P | aHR (95%CI) | P |
|---|---|---|---|---|---|---|---|---|---|---|
| | No | | | Yes | | | | | | |
| | Event | PY | IR | Event | PY | IR | | | | |
| All-cause mortality | 57 | 11745 | 0.5 | 37 | 12233 | 0.3 | 0.62 (0.41,0.94) | 0.02 | 0.6 (0.39,0.92) | 0.02 |
| CV death | 22 | 11745 | 0.2 | 16 | 12233 | 0.1 | 0.7 (0.37,1.33) | 0.27 | 0.61 (0.31,1.19) | 0.15 |
| Hospitalized CAD | 88 | 11480 | 0.8 | 94 | 11955 | 0.8 | 1.02 (0.76,1.37) | 0.89 | 1.01 (0.75,1.35) | 0.97 |
| Hospitalized stroke | 118 | 11326 | 1.0 | 74 | 12019 | 0.6 | 0.59 (0.44,0.79) | <.001 | 0.52 (0.39,0.7) | <.001 |
| Ischemic stroke | 64 | 11326 | 0.57 | 32 | 12019 | 0.27 | 0.47 (0.31,0.72) | <.001 | 0.41 (0.27,0.64) | <.001 |
| Hemorrhagic stroke | 15 | 11326 | 0.13 | 15 | 12019 | 0.12 | 0.94 (0.46,1.93) | 0.88 | 0.88 (0.42,1.83) | 0.72 |
| Hospitalized heart failure | 31 | 11688 | 0.3 | 32 | 12147 | 0.3 | 0.99 (0.6,1.62) | 0.96 | 0.91 (0.55,1.52) | 0.72 |

IR, incidence rate, per 100 person-years; CI, confidence interval; *p*, *p* value; cHR, crude hazard ratio; aHR, adjusted hazard ratio, controlling for sex, age, area, every comorbidity, and drug in Table 1.

lower in ULT users than in ULT nonusers. We also assessed the composite outcomes of all-cause mortality and hospitalized CAD, stroke, and HF between ULT users and nonusers; the IR of ULT users vs nonusers was 1.2 vs 1.6 (aHR = 0.52, 95% CI = 0.55–0.86, $p < 0.001$).

The subgroup analysis of hospitalized stroke (S1 Table) revealed that patients taking uricosuric agents; patients with underlying HT or stroke without underlying DM, CAD, stroke, or HF; and patients using ACE inhibitors, ARBs, β-blockers, calcium channel blockers, or diuretics could have a significantly low risk of hospitalized stroke. The subgroup analysis of all-cause mortality (S2 Table) indicated that patients taking xanthine oxidase (XO) inhibitors; male patients; patients with underlying stroke without underlying HT, DM, CAD, or hypercholesterolemia; and patients using β-blockers or diuretics not using ACE inhibitors, ARBs, or calcium channel blockers could demonstrate a significantly low risk of all-cause mortality.

The cumulative incidence of hospitalized stroke and all-cause mortality in ULT users and nonusers, obtained using the Kaplan–Meier method, is presented in Fig 2. The *p* values obtained from the log-rank test were <0.001 and 0.02 for hospitalized stroke and all-cause mortality, respectively.

Uricosuric agent users with a cumulative therapy duration of ≤1, 1–5, and >5 months were, respectively, 0.5, 0.56, and 0.31 times less likely to develop hospitalized stroke compared with nonusers, as presented in S3 Table (*p* for trend < 0.001). Uricosuric agent users with cumulative mean DDDs of (per month) ≤0.5, 0.5–0.8, and >0.8 were 0.45, 0.38, and 0.39 times less likely, respectively, to experience hospitalized stroke than nonusers (*p* for trend < 0.01).

## Discussion

Our study demonstrated that ULT was associated with lower risks of hospitalized stroke and all-cause mortality in patients with gout. Subgroup analyses also revealed that uricosuric agents and XO inhibitors were associated with lower risks of hospitalized stroke and decreased risk of mortality, respectively. A dose–response trend was observed in the effects of uricosuric agents on the decrease in hospitalized stroke risk. Recent literature has demonstrated that SUA plays a clear and independent role in the development of CV diseases. First, SUA was demonstrated to stimulate platelet-derived growth factor receptor-β (PDGFR-β) phosphorylation through the mitogen-activated protein kinase pathway and to induce vascular smooth muscle cell proliferation in cultured rat aortic cells [17]. Hyperuricemia, which is associated with endothelial cell dysfunction and increased reactive oxygen species production, can increase senescence

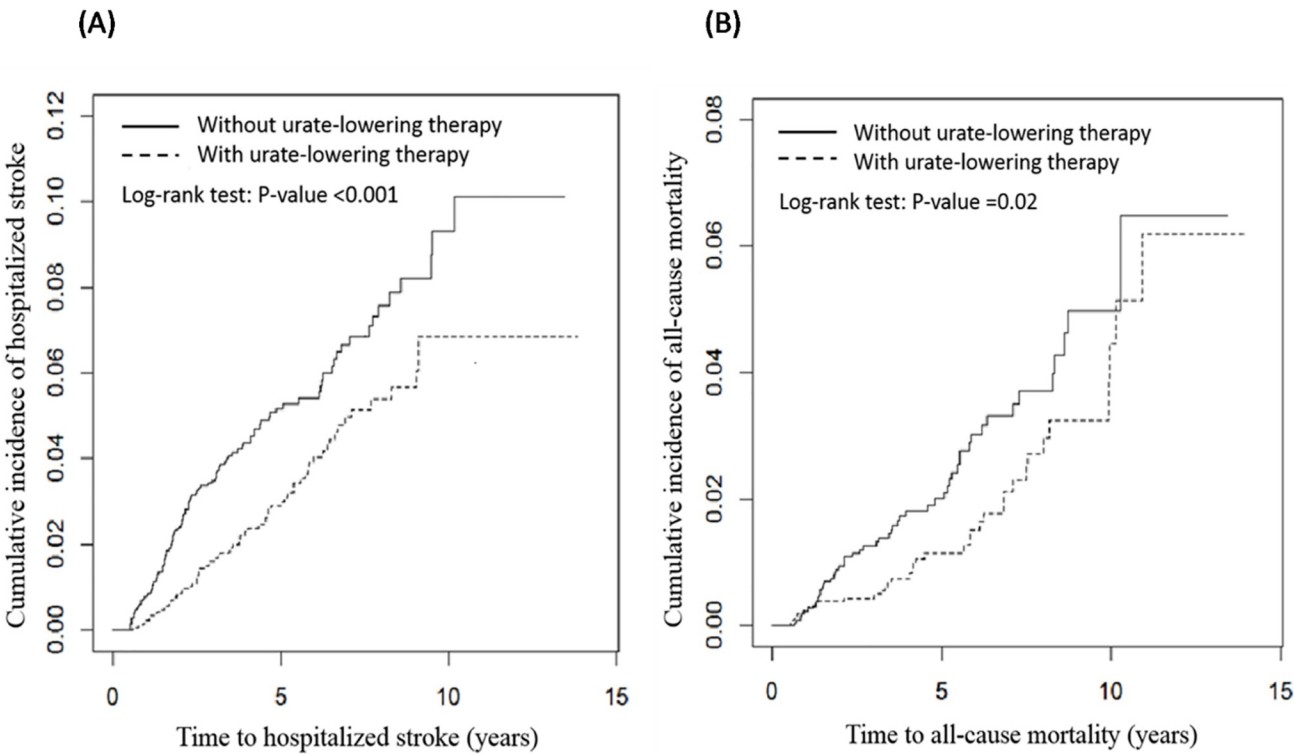

**Fig 2. Cumulative incidence of (A) hospitalized stroke and (B) all-cause mortality in patients receiving and not receiving urate-lowering therapy.**

and apoptosis of vascular cells, resulting in accelerated atherosclerosis [18]. Second, UA increases the circulating levels of inflammatory mediators, and these proinflammatory effects of UA add to its proatherogenic properties [19]. Third, gout, characterized by intermittent inflammatory attacks, also increases the risks of morbidity and mortality from CV diseases [20].

XO inhibitors [12, 13, 14, 21, 22] were reported to be associated with decreased risks of composite CV and mortality outcomes, and our results are consistent with these findings. In our study, ULT was significantly associated with lower risks of composite outcomes, that is, all-cause mortality and hospitalized CAD, stroke, and HF, in patients with gout. A study revealed that SUA was an independent predictor of stroke or excess mortality in patients with DM, HT, and atrial fibrillation; elderly persons; and the general population [5]. It has also been demonstrated to be related to carotid atherosclerosis [23], arterial wave reflection [24], and intercellular adhesion molecule-1 levels [25]. These data suggest the development or progression of stroke may be attenuated by ULT. Our study demonstrated that ULT was associated with a reduced risk of hospitalized stroke, and this benefit is mostly driven by the diminution of ischemic stroke. Uricosuric agents are more effective than XO inhibitors in decreasing hospitalized stroke, which may indicate that this protective effect might be due to increasing SUA excretion and may not be due to the decrease in XO activity. ULT has been reported to reduce blood pressure [8, 9], with a greater impact on stroke than on heart disease [26]. This may explain our result of ULT being associated with a low risk of stroke and with no significant risk of CAD and heart failure.

The causal relationship of SUA and all-cause mortality is controversial. The Framingham Heart Study did not report a significant association between SUA and all-cause mortality [6],

whereas other studies have revealed that a high level of SUA significantly increased the risk of all-cause mortality [27], and that allopurinol use significantly reduced mortality [12, 21, 28, 29]. In our study, ULT users showed a lower risk of all-cause mortality than the nonusers. The subgroup analysis disclosed that XO inhibitors were more strongly associated with a lower risk of death than uricosuric agents, which might indicate that this protective effect occurs through the reduction of XO activity and pro-oxidants and not solely with the increased excretion of UA. A meta-analysis reported that hyperuricemia marginally increased the risk of CAD events [2]. Norman et al. disclosed that high-dose allopurinol significantly prolonged the time to ST depression in patients with stable angina [30]. However, allopurinol users demonstrated a decrease [31] or no significant change [32] in the risk of myocardial infarction compared with nonusers. No significant differences were observed in hospitalized CAD and CV death between ULT users and nonusers in our study. Therefore, whether SUA is a risk factor for the development or progression of CAD remains unclear. One meta-analysis reported that an increase of SUA by 1 mg/dL increased the odds of HF by 19% [4]. Allopurinol has been suggested to improve a range of surrogate HF markers [33]; however, a clinical study failed to observe the therapeutic effects of allopurinol in patients with HF [34]. No significant differences were observed in hospitalized HF between ULT users and nonusers in our study. However, the IR of hospitalized HF was low in our study, which might be explained by the relatively young age of our study sample.

The strengths and limitations of our study warrant discussion. First, randomized clinical trials with long-term follow-up were not eligible and unavailable to answer our study questions. We therefore performed a large series cohort study with propensity score matching of 30 clinical variables to maximally balance the possible confounders. However, physicians may not prescribe ULT for patients with milder gout or fewer recurrent attacks of gout. Some patients also may not be followed up and may not take UA tests or adhere to the ULT regimen. These confounders may influence our outcomes. Second, the database does not contain information on smoking status, alcohol consumption, physical activity, and body mass index of patients, which might have influenced the results of our censored outcomes. Third, we could not obtain the blood pressure levels of patients or their biochemical blood test results, such as UA, blood glucose, hemoglobin A1C, cholesterol and creatinine. Patients with lower UA levels may not require ULT. Without confirmation of UA or crystal identification, other forms of arthritis may be misdiagnosed as gout. These potential biases also require attention. Fourth, the cumulative duration and mean dosage of ULT were short and low, respectively. This might be explained by the physicians' poor adherence to treatment guidelines for gout or the poor adherence of patients to hypouricemic agent use. Fifth, our results may not be applicable to hyperuricemic patients without attacks of gout because we selected patients with gout. Finally, our study was a retrospective cohort study with inevitable existing biases; a randomized control study is required to confirm our results.

Thus, our study results revealed that ULT was associated with low risks of hospitalized stroke and all-cause mortality. Uricosuric agents seemed to reduce the risk of admitted stroke in a dose–response manner. Patients with gout may need to be on constant ULT to attenuate the risks of stroke and death.

## Supporting information

**S1 Table. Incidence and hazard ratio of hospitalized stroke between cohorts receiving or not receiving urate-lowering therapy.**
(DOCX)

**S2 Table. Incidence and hazard ratio of all-cause mortality between cohorts receiving or not receiving urate-lowering therapy.**
(DOCX)

**S3 Table. Cox model measured hazard ratios of hospitalized stroke associated with variables restricted on uricosuric agents.**
(DOCX)

## Acknowledgments

This manuscript was edited by Wallace Academic Editing.

## Author Contributions

**Conceptualization:** Fu-Shun Yen, Chii-Min Hwu.

**Data curation:** Hsin-Lun Li, James Cheng-Chung Wei.

**Formal analysis:** Chih-Cheng Hsu, Hsin-Lun Li.

**Writing – original draft:** Fu-Shun Yen, Hsin-Lun Li.

**Writing – review & editing:** Chih-Cheng Hsu, James Cheng-Chung Wei, Chii-Min Hwu.

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
