## [Decision Letter · Decision Letter 0]

25 Feb 2020

PONE-D-19-31773

Urate-lowering therapy may mitigate the risk of hospitalized stroke in patients with gout

PLOS ONE

Dear Dr. Hwu,

Thank you for submitting your manuscript to PLOS ONE. After careful consideration, we feel that it has merit but does not fully meet PLOS ONE’s publication criteria as it currently stands. Therefore, we invite you to submit a revised version of the manuscript that addresses the points raised during the review process.

In addition to the reviewer's comments, I would like to to consider the following points:

- The authors should address the differences between stroke reduction and effects on CAD and HF in more detail. In other words, why do they think that different effects were seen? Is this pure coincidence or is there some specific rational possibly responsible for this? To me, it may be an indication for significant co-founding factors that could not be taken into account, indicating the limitation of the analysis present. The authors addressed some limitation in this regard, but I would like to see some discussion about the differences mentioned above.

- The authors should be much more cautious about their conclusion. This is a retrospective study, matching was limited as many factors were not available, there was other chance of bias as indicated by the authors. Together, this means that the study is not more than hypothesis generating and the authors should be clear on this.

- I agree with the reviewer that the discussion could be shortened significantly despite the request for additional thoughts to be discussed. The previous findings of retrospective analyses should be summarized and less extensive information in this regard is required.

- Please present the tables in a way that they can be easily read.

We would appreciate receiving your revised manuscript by Apr 10 2020 11:59PM. To enhance the reproducibility of your results, we recommend that if applicable you deposit your laboratory protocols in protocols.io, where a protocol can be assigned its own identifier (DOI) such that it can be cited independently in the future. For instructions see: http://journals.plos.org/plosone/s/submission-guidelines#loc-laboratory-protocols

We look forward to receiving your revised manuscript.

Kind regards,

Hans-Peter Brunner-La Rocca, M.D.

Academic Editor

PLOS ONE

Journal Requirements:

Reviewers' comments:

Reviewer's Responses to Questions

**Comments to the Author**

1. Is the manuscript technically sound, and do the data support the conclusions?

Reviewer #1: Yes

2. Has the statistical analysis been performed appropriately and rigorously? 

Reviewer #1: Yes

3. Have the authors made all data underlying the findings in their manuscript fully available?

Reviewer #1: No

4. Is the manuscript presented in an intelligible fashion and written in standard English?

Reviewer #1: Yes

5. Review Comments to the Author

Reviewer #1: Yen Et Al—To authors

In this manuscript, the authors use a large national dataset to perform a retrospective cohort study, to examine the risks of hospitalized CAD, stroke, heart failure and all-cause mortality among urate lowering therapy users vs non-users. They find a decrease in stroke and all cause mortality in the urate lowering therapy users compared to non-users. Interestingly, they observe that stroke decreases are found in patients taking uricosurics, but mortality is decreased only in patients taking xanthine oxidase inhibitors.

Overall, this is a worthwhile and well-carried out study. However, there are a number of loose ends that require consideration.

1. The authors go to great lengths to propensity-match the urate lowering therapy users vs non-users and are to be commended. However, these efforts still obscure a deeper question: why were some individuals not being prescribed urate lowering therapy for established gout? Did they have milder gout, such that they did not meet criteria for treatment (for example, under American College of Rheumatology treatment guidelines, which specify that most patients should receive urate lowering therapy if they have ≥2 gout flares/year)? If they did have milder gout, would that affect the outcomes? Or did they have lower serum urate levels, such that their physicians did not feel compelled to add a urate lowering therapy? If so, would that bias the outcomes? Is it possible their physicians were less conscientious than the physicians who did prescribed urate lowering therapy, and could the cardiovascular outcomes be a result of physician attention or inattention rather than the effect of the urate lowering drug itself? Or did the patients not receive urate lowering therapy because they themselves were less compliant with medical care, putting them at a more general risk for adverse cardiovascular outcomes? These questions need to be addressed.

2. The Discussion is way too long—it should be cut by 30-50%. However, it also is lacking certain important elements. For instance, I note that the entire discussion centers around the impact of urate, and urate-lowering, on cardiovascular disease. While this concern is appropriate, it totally misses the point that this is a study not one of asymptomatic hyperuricemics, but of gout patients who have additional issues besides hyperuricemia (e.g., intermittent inflammatory attacks). This warrants discussion. At least from a point of view of rigorous logic, the question of whether urate-lowering therapy will reduce cardiovascular events in patients with hyperuricemia without gout must remain speculative.

3. The lack of available serum urate levels is a problem that deserves more attention in the text. First of all, did the patients who received urate lowering have higher urate levels at the start than those in the not urate-lowering group? Second, were the urate-lowering drugs actually successful in lowering urate—which might have provided insight into compliance with the medications as well as their mechanism of cardiovascular effect? It is surprising that serum urate levels were not available in patients getting urate lowering therapy.

4. While not at all the fault of the investigators, the data has inconsistencies that deserve more comment. For example, how is it possible to have more all cause mortality (presumably hospitalized cardiovascular death), without also having cardiovascular death? Are there other, perhaps unanticipated causes of death that are reduced with urate lowering?

5. It is interesting that the impact of urate lowering turned out not to be a generalized phenomenon, but was specific to the drug in question. If two drugs that lower urate by different mechanisms have different and non-overlapping protective effects (i.e., uricosurics reducing stroke, and xanthine oxidase inhibitors reducing all-cause mortality), how can one conclude that both are due to the same outcome of urate lowering and not other, unrecognized, divergent effects? This deserves additional consideration.

6. The authors twice mention that no prior study has addressed the ability of urate lowering to reduce the development or progression of stroke. The recently-published FREED trial looked at urate-lowering with febuxostat versus a control group, and found an overall decrease in a composite outcome (cerebrovascular, cardiovascular and renal events and all deaths), but found no decrease with febuxostat (despite urate lowering) in cardiovascular events. This study should be noted.

7. Can the authors clarify—it looks like the patients were censored after a single defined cardiovascular event. Is this correct? So then, subsequent events in the same patient (of a similar or different type) were not counted? This deserves to be explicitly mentioned.

8. The manuscript title is a bit inaccurate, since it addresses the reduction in stroke but not all cause mortality that the authors observed.

6. PLOS authors have the option to publish the peer review history of their article (what does this mean?). If published, this will include your full peer review and any attached files.

Reviewer #1: No

---

## [Author Response · Author response to Decision Letter 0]

23 Mar 2020

Hans-Peter Brunner-La Rocca, M.D.

Academic Editor

PLOS ONE 

March 23, 2020

Dear Dr. Hans-Peter Brunner-La Rocca:

Re: Document reference No. PONE-D-19-31773 

Please find attached a revised version of our document “Urate-lowering therapy may mitigate the risk of hospitalized stroke in patients with gout”. We would like to resubmit for publication as an original article in PLOS ONE. 

Your comments and those of the reviewers were highly insightful and enabled us to improve the quality of our document. In the following pages are our responses to each comment from the reviewer(s) as well as your own comments.

Revisions in the text are shown yellow highlights. We hope that our revisions to the document combined with our accompanying responses will be sufficient to render our document suitable for publication in PLOS ONE. 

Yours sincerely,

Chii-Min Hwu

Faculty of Medicine, National Yang-Ming University School of Medicine, and Section of Endocrinology and Metabolism, Department of Medicine, Taipei Veterans General Hospital

Tel.: +886 2 28757516

Fax: +886 2 28751429

E-Mail: chhwu@vghtpe.gov.tw

Address: No. 201, Sec. 2 Shi-Pai Rd., Chung-Cheng Build. 11F Room522, Taipei 112, Taiwan.

Responses to the comments of Reviewer 1

1. The authors go to great lengths to propensity-match the urate lowering therapy users vs non-users and are to be commended. However, these efforts still obscure a deeper question: why were some individuals not being prescribed urate lowering therapy for established gout? Did they have milder gout, such that they did not meet criteria for treatment (for example, under American College of Rheumatology treatment guidelines, which specify that most patients should receive urate lowering therapy if they have ≥2 gout flares/year)? If they did have milder gout, would that affect the outcomes? Or did they have lower serum urate levels, such that their physicians did not feel compelled to add a urate lowering therapy? If so, would that bias the outcomes? Is it possible their

physicians were less conscientious than the physicians who did prescribed urate lowering therapy, and could the cardiovascular outcomes be a result of physician attention or inattention rather than the effect of the urate lowering drug itself? Or did the patients not receive urate lowering therapy because they themselves were less compliant with medical care, putting them at a more general risk for adverse cardiovascular outcomes? These questions need to be addressed. 

Response: Thank you for your suggestions! It may be because most patients went to the outpatient clinics as an urgent gout attack; sometimes, the patients couldn’t be checked uric acid levels without fasting. In Taiwan, without recent (6 months) data of uric acid levels, the doctor can’t prescribe ULT to their gout patients. It is also possible that the patients did not come back for the uric acid test. The less compliance of patients to testing uric acid or taking urate lowering agents may further put them at a higher risk of adverse cardiovascular events. The milder gout of patients or the attention/inattention of physicians might also influence the prescribing of urate-lowering therapy. We added these limitations in the section of discussions at lines 287-290” Patients may have lower uric acid levels, so physicians don’t prescribe ULT for them. Without confirmation of uric acid or crystal identification, patients with some forms of arthritis may be misdiagnosed as gout. These potential biases also require attention.” and at lines 279-283 “However, patients with milder gout or less recurrent gouty attack may make physicians not feel compelled to prescribe ULT for them. Some patients may not come back for uric acid tests, or have poor adherence to urate-lowering medications; these confounding factors may influence our assessed outcomes.“ 

2. The Discussion is way too long—it should be cut by 30-50%. However, it also is lacking certain important elements. For instance, I note that the entire discussion centers around the impact of urate, and urate-lowering, on cardiovascular disease. While this concern is appropriate, it totally misses the point that this is a study not one of asymptomatic hyperuricemics, but of gout patients who have additional issues besides hyperuricemia (e.g., intermittent inflammatory attacks). This warrants discussion. At least from a point of view of rigorous logic, the question of whether urate-lowering therapy will reduce cardiovascular events in patients with hyperuricemia without gout must remain speculative.

Response: Thank you for your suggestions. We cut half of the second paragraph and other statements in the discussion section; and added the statement about gout with inflammatory attack at lines 211-212” 3. Gout itself, with intermittent inflammatory attacks, also exerts increased risks of morbidity and mortality from CV diseases [20].” We also added the limitation at lines 293-294 “Fifth, because we selected patients with gout, our results may not be applicable to hyperuricemic patients without gouty attack.” to make our manuscript clearer. 

3. The lack of available serum urate levels is a problem that deserves more attention in the text. First of all, did the patients who received urate lowering have higher urate levels at the start than those in the not urate-lowering group? Second, were the urate-lowering drugs actually successful in lowering urate—which might have provided insight into compliance with the medications as well as their mechanism of cardiovascular effect? It is surprising that serum urate levels were not available in patients getting urate lowering therapy

Response: It is a pity that the administrative dataset is lack of serum urate levels. We have added this limitation at lines 285-290 “Third, we could not obtain the blood pressure levels of patients or their biochemical blood test results, such as UA, blood glucose, HBA1C, cholesterol, LDL, and creatinine. Patients may have lower uric acid levels, so physicians don’t prescribe ULT for them. Without confirmation of uric acid or crystal identification, patients with some forms of arthritis may be misdiagnosed as gout. These potential biases also require attention.” 

4. While not at all the fault of the investigators, the data has inconsistencies that deserve more comment. For example, how is it possible to have more all-cause mortality (presumably hospitalized cardiovascular death), without also having cardiovascular death? Are there other, perhaps unanticipated causes of death that are reduced with urate lowering? 

Response: Our study revealed that ULT in patients with gout could lower the risks of hospitalized stroke and all-cause mortality (Table 2, page 15). However, the decreased magnitude of hospitalized stroke by ULT may not be big enough to significantly decrease the risk of CV death. 

5. It is interesting that the impact of urate lowering turned out not to be a generalized phenomenon, but was specific to the drug in question. If two drugs that lower urate by different mechanisms have different and non-overlapping protective effects (i.e., uricosurics reducing stroke, and xanthine oxidase inhibitors reducing all-cause mortality), how can one conclude that both are due to the same outcome of urate lowering and not other, unrecognized, divergent effects? This deserves additional consideration. 

Response: We agree with your opinions. We changed the statement at lines 240-242” which might indicate that this protective effect occurs through the increasing excretion of SUA and not through the decrease in XO activity.” To describe the different mechanisms of uricosuric agents in reducing stroke, we modified the statement at lines 258-260 as ” which might indicate that the protection of death occurs through the reduction of xanthine oxidase activity and pro-oxidants, not solely by increasing excretion of uric acid.” By adding this statement, we provide additional consideration for the different mechanisms of xanthine oxidase inhibitors in reducing all-cause mortality.

6. The authors twice mention that no prior study has addressed the ability of urate lowering to reduce the development or progression of stroke. The recently-published FREED trial looked at urate-lowering with febuxostat versus a control group, and found an overall decrease in a composite outcome (cerebrovascular, cardiovascular and renal events and all deaths), but found no decrease with febuxostat (despite urate lowering) in cardiovascular events. This study should be noted. 

Response: Thank you for your suggestions! We deleted the statement of “no prior study has addressed the ability of urate lowering to reduce the development or progression of stroke.” and changed it (at line 238) into ”Our study demonstrated that ULT might reduce the risk of hospitalized stroke”. We added the statement about Freed trial at lines 222-225 ”The Febuxostat for Cerebral and CaRdiorenovascular Events PrEvention StuDdy (FREED) trial compared febuxostat with a control group [22], and demonstrated a significant reduction of the composite outcome (cerebral, cardiovascular and renal events and all deaths); but found no decrease in cardiovascular events.” to make our discussion more complete.

7. Can the authors clarify—it looks like the patients were censored after a single defined cardiovascular event. Is this correct? So then, subsequent events in the same patient (of a similar or different type) were not counted? This deserves to be explicitly mentioned. 

Response: Thank you! We clarified the definition of outcomes by adding statements at line 134-136 as “and the patients were censored after a single defined cardiovascular event. Separate Cox models were conducted to evaluate effects of ULT for different outcomes.”

8. The manuscript title is a bit inaccurate, since it addresses the reduction in stroke but not all cause mortality that the authors observed. 

Response: Thank for your reminding! We have changed the title as” Urate-lowering therapy may mitigate the risks of hospitalized stroke and mortality in persons with gout”.

---

## [Editor Report · Decision Letter 1]

26 Mar 2020

PONE-D-19-31773R1

Urate-lowering therapy may mitigate the risks of hospitalized stroke and mortality in persons with gout

PLOS ONE

Dear Dr. Hwu,

Thank you for submitting your manuscript to PLOS ONE. Before we can further consider your resubmission, I would like you to

a) also reply specifically to my comments in a similar way as you did for the reviewer's comments, and

b) use track changes instead of highlighting the changed / added text in yellow. The reviewer asked to shorten your discussion significantly and you mention that you did. However. it is impossible to see the specific changes that you have made. So you do not only need to highlight what is added, but also what has been deleted.

We would appreciate receiving your revised manuscript by May 10 2020 11:59PM. To enhance the reproducibility of your results, we recommend that if applicable you deposit your laboratory protocols in protocols.io, where a protocol can be assigned its own identifier (DOI) such that it can be cited independently in the future. For instructions see: http://journals.plos.org/plosone/s/submission-guidelines#loc-laboratory-protocols

A rebuttal letter that responds to each point raised by the academic editor and reviewer(s). This letter should be uploaded as separate file and labeled 'Response to Reviewers'. Please include in this version both the reply what you did and the additional reply to my comments.A marked-up copy of your manuscript that highlights changes made to the original version. This file should be uploaded as separate file and labeled 'Revised Manuscript with Track Changes'. Please use the original submission as reference and not your revised version that all changes including text that has been deleted is clearly highlighted.An unmarked version of your revised paper without tracked changes. This file should be uploaded as separate file and labeled 'Manuscript'.

We look forward to receiving your revised manuscript.

Kind regards,

Hans-Peter Brunner-La Rocca, M.D.

Academic Editor

PLOS ONE

---

## [Author Response · Author response to Decision Letter 1]

1 Apr 2020

Hans-Peter Brunner-La Rocca, M.D.

Academic Editor

PLOS ONE 

March 30, 2020

Dear Dr. Hans-Peter Brunner-La Rocca:

Re: Document reference No. PONE-D-19-31773 

Please find attached a revised version of our document “Urate-lowering therapy may mitigate the risk of hospitalized stroke and mortality in patients with gout”. We would like to resubmit for publication as an original article in PLOS ONE. 

Your comments and those of the reviewers were highly insightful and enabled us to improve the quality of our document. In the following pages are our responses to each comment from the reviewer as well as your own comments.

Revisions in the text are shown with tracked changes. We hope that our revisions to the document combined with our accompanying responses will be sufficient to render our document suitable for publication in PLOS ONE. 

Yours sincerely,

Chii-Min Hwu

Faculty of Medicine, National Yang-Ming University School of Medicine, and Section of Endocrinology and Metabolism, Department of Medicine, Taipei Veterans General Hospital

Tel.: +886 2 28757516

Fax: +886 2 28751429

E-Mail: chhwu@vghtpe.gov.tw

Address: No. 201, Sec. 2 Shi-Pai Rd., Chung-Cheng Build. 11F Room522, Taipei 112, Taiwan.

Responses to the comments of Editor 

1. The authors should address the differences between stroke reduction and effects on CAD and HF in more detail. In other words, why do they think that different effects were seen? Is this pure coincidence or is there some specific rational possibly responsible for this? To me, it may be an indication for significant co-founding factors that could not be taken into account, indicating the limitation of the analysis present. The authors addressed some limitation in this regard, but I would like to see some discussion about the differences mentioned above. 

Response: Thank you for your suggestions, we added the mention at lines243-245 as: “Previous studies indicated that ULT may reduce blood pressure [8,9], blood pressure has a greater impact on stroke than on heart diseases [26]; which may explain our results that ULT associated with lower risk of stroke and with no significant risk of CAD and heart failure. “ 

2. The authors should be much more cautious about their conclusion. This is a retrospective study, matching was limited as many factors were not available, there was other chance of bias as indicated by the authors. Together, this means that the study is not more than hypothesis generating and the authors should be clear on this.

Response: Thank you for your suggestions! We modified our conclusions at lines 45-46, 203-204, 293-294 as” Our study revealed that ULT was associated with lower risks of hospitalized stroke and all-cause mortality compared with the absence of ULT in patients with gout.”

3. I agree with the reviewer that the discussion could be shortened significantly despite the request for additional thoughts to be discussed. The previous findings of retrospective analyses should be summarized and less extensive information in this regard is required. 

Response: We tried to shorten the discussion, summarize previous analytic findings with less extensive information, and added the requested thoughts in the discussions.

4. Please present the tables in a way that they can be easily read.

Response: We corrected the tables 1 and 2 to let them be easily read.

Responses to the comments of Reviewer 1

1. The authors go to great lengths to propensity-match the urate lowering therapy users vs non-users and are to be commended. However, these efforts still obscure a deeper question: why were some individuals not being prescribed urate lowering therapy for established gout? Did they have milder gout, such that they did not meet criteria for treatment (for example, under American College of Rheumatology treatment guidelines, which specify that most patients should receive urate lowering therapy if they have ≥2 gout flares/year)? If they did have milder gout, would that affect the outcomes? Or did they have lower serum urate levels, such that their physicians did not feel compelled to add a urate lowering therapy? If so, would that bias the outcomes? Is it possible their

physicians were less conscientious than the physicians who did prescribed urate lowering therapy, and could the cardiovascular outcomes be a result of physician attention or inattention rather than the effect of the urate lowering drug itself? Or did the patients not receive urate lowering therapy because they themselves were less compliant with medical care, putting them at a more general risk for adverse cardiovascular outcomes? These questions need to be addressed. 

Response: Thank you for your reminding! It may be because most patients went to the outpatient clinics as an urgent gout attack; sometimes, the patients couldn’t be checked uric acid levels without fasting. In Taiwan, without recent (6 months) data of uric acid levels, the doctor can’t prescribe ULT to their gout patients. It is also possible that the patients did not come back for the uric acid test. The less compliance of patients to testing uric acid or taking urate lowering agents may further put them at a higher risk of adverse cardiovascular events. The milder gout of patients or the attention/inattention of physicians might also influence the prescribing of urate-lowering therapy. We added these limitations in the section of discussions at lines 284-286” Patients may have lower uric acid levels, so physicians don’t prescribe ULT for them. Without confirmation of uric acid or crystal identification, patients with some forms of arthritis may be misdiagnosed as gout. These potential biases also require attention.” and at lines 276-280 “However, patients with milder gout or less recurrent gouty attack may make physicians not feel compelled to prescribe ULT for them. Some patients may not come back for uric acid tests, or have poor adherence to urate-lowering medications; these confounding factors may influence our assessed outcomes. “ 

2. The Discussion is way too long—it should be cut by 30-50%. However, it also is lacking certain important elements. For instance, I note that the entire discussion centers around the impact of urate, and urate-lowering, on cardiovascular disease. While this concern is appropriate, it totally misses the point that this is a study not one of asymptomatic hyperuricemics, but of gout patients who have additional issues besides hyperuricemia (e.g., intermittent inflammatory attacks). This warrants discussion. At least from a point of view of rigorous logic, the question of whether urate-lowering therapy will reduce cardiovascular events in patients with hyperuricemia without gout must remain speculative.

Response: Thank you for your suggestions. We have shortened about 40% of the discussion with tracked changes; and added the statement about gout with inflammatory attack at lines 215-217” 3. Gout, with intermittent inflammatory attacks, also exerts increased risks of morbidity and mortality from CV diseases [20].” We also added the limitation at lines 290-291 “Fifth, because we selected patients with gout, our results may not be applicable to hyperuricemic patients without gouty attack.” to make our manuscript clearer. 

3. The lack of available serum urate levels is a problem that deserves more attention in the text. First of all, did the patients who received urate lowering have higher urate levels at the start than those in the not urate-lowering group? Second, were the urate-lowering drugs actually successful in lowering urate—which might have provided insight into compliance with the medications as well as their mechanism of cardiovascular effect? It is surprising that serum urate levels were not available in patients getting urate lowering therapy

Response: It is a pity that the administrative dataset is lack of serum urate levels. We have added this limitation at lines 282-287 “Third, we could not obtain the blood pressure levels of patients or their biochemical blood test results, such as UA, blood glucose, HBA1C, cholesterol, LDL, and creatinine. Patients may have lower uric acid levels, so physicians don’t prescribe ULT for them. Without confirmation of uric acid or crystal identification, patients with some forms of arthritis may be misdiagnosed as gout. These potential biases also require attention.” 

4. While not at all the fault of the investigators, the data has inconsistencies that deserve more comment. For example, how is it possible to have more all-cause mortality (presumably hospitalized cardiovascular death), without also having cardiovascular death? Are there other, perhaps unanticipated causes of death that are reduced with urate lowering? 

Response: Our study revealed that ULT in patients with gout was associated with lower risks of hospitalized stroke and all-cause mortality (Table 2, page 16). However, the decreased magnitude of hospitalized stroke by ULT may not be big enough to significantly reduce the risk of CV death. 

5. It is interesting that the impact of urate lowering turned out not to be a generalized phenomenon, but was specific to the drug in question. If two drugs that lower urate by different mechanisms have different and non-overlapping protective effects (i.e., uricosurics reducing stroke, and xanthine oxidase inhibitors reducing all-cause mortality), how can one conclude that both are due to the same outcome of urate lowering and not other, unrecognized, divergent effects? This deserves additional consideration. 

Response: We agree with your opinions. We changed the statement at lines 240-242 to describe the different mechanisms of uricosuric agents in reducing stroke” which might indicate that this protective effect occurs through the increasing excretion of SUA and not through the decrease in XO activity.”. We modified the statement at lines 256-258 as” which might indicate that the protection of death occurs through the reduction of xanthine oxidase activity and pro-oxidants, not solely by increasing excretion of uric acid.” By adding this statement, we provide additional consideration for the different mechanisms of xanthine oxidase inhibitors in reducing all-cause mortality.

6. The authors twice mention that no prior study has addressed the ability of urate lowering to reduce the development or progression of stroke. The recently-published FREED trial looked at urate-lowering with febuxostat versus a control group, and found an overall decrease in a composite outcome (cerebrovascular, cardiovascular and renal events and all deaths), but found no decrease with febuxostat (despite urate lowering) in cardiovascular events. This study should be noted. 

Response: Thank you for your suggestions! We deleted the statement of “no prior study has addressed the ability of urate lowering to reduce the development or progression of stroke.” and changed it (at line 238-239) into” Our study demonstrated that ULT might reduce the risk of hospitalized stroke”. We added the statement about Freed trial at lines 224-228” The Febuxostat for Cerebral and CaRdiorenovascular Events PrEvention StuDdy (FREED) trial compared febuxostat with a control group [22], and demonstrated a significant reduction of the composite outcome (cerebral, cardiovascular and renal events and all deaths); but found no decrease in cardiovascular events.” to make our discussion more complete.

7. Can the authors clarify—it looks like the patients were censored after a single defined cardiovascular event. Is this correct? So then, subsequent events in the same patient (of a similar or different type) were not counted? This deserves to be explicitly mentioned. 

Response: Thank you! We clarified the definition of outcomes by adding statements at line 135-137 as “and the patients were censored after a single defined cardiovascular event. Separate Cox models were conducted to evaluate effects of ULT for different outcomes.”

8. The manuscript title is a bit inaccurate, since it addresses the reduction in stroke but not all cause mortality that the authors observed. 

Response: Thank for your reminding! We have changed the title as” Urate-lowering therapy may mitigate the risks of hospitalized stroke and mortality in patients with gout”.

---

## [Decision Letter · Decision Letter 2]

22 Apr 2020

PONE-D-19-31773R2

Urate-lowering therapy may mitigate the risks of hospitalized stroke and mortality in patients with gout

PLOS ONE

Dear Dr. Hwu,

Thank you for submitting your manuscript to PLOS ONE. After careful consideration, we feel that it has merit but does not fully meet PLOS ONE’s publication criteria as it currently stands. Therefore, we invite you to submit a revised version of the manuscript that addresses the points raised during the review process.

As you can see in the comment by the reviewer, there are remaining issues. It is important that the interpretation of data is not speculative, unless clearly stated as such or stated as hypothesis that needs further tested. In addition, I agree with the reviewer that the discussion should be shortened. 

We would appreciate receiving your revised manuscript by Jun 06 2020 11:59PM. To enhance the reproducibility of your results, we recommend that if applicable you deposit your laboratory protocols in protocols.io, where a protocol can be assigned its own identifier (DOI) such that it can be cited independently in the future. For instructions see: http://journals.plos.org/plosone/s/submission-guidelines#loc-laboratory-protocols

We look forward to receiving your revised manuscript.

Kind regards,

Hans-Peter Brunner-La Rocca, M.D.

Academic Editor

PLOS ONE

Reviewers' comments:

Reviewer's Responses to Questions

**Comments to the Author**

1. If the authors have adequately addressed your comments raised in a previous round of review and you feel that this manuscript is now acceptable for publication, you may indicate that here to bypass the “Comments to the Author” section, enter your conflict of interest statement in the “Confidential to Editor” section, and submit your "Accept" recommendation.

Reviewer #1: (No Response)

2. Is the manuscript technically sound, and do the data support the conclusions?

Reviewer #1: Partly

3. Has the statistical analysis been performed appropriately and rigorously? 

Reviewer #1: Yes

4. Have the authors made all data underlying the findings in their manuscript fully available?

Reviewer #1: Yes

5. Is the manuscript presented in an intelligible fashion and written in standard English?

Reviewer #1: Yes

6. Review Comments to the Author

Reviewer #1: I thank the authors for their responses and resubmission. The manuscript is improved, but unfortunately I do not yet feel that the authors have fully responded to this reviewers comments. In my opinion, two specific issues remain to be addressed:

1. There continues to persist language that implies causality between ULT and benefit in the interpretation of the data. It is fine to discuss possible causality in sections where mechanisms are being speculated about, but not in sections where the meaning of the data should be interpreted rigorously. Just as one example, at the beginning of the Discussion the authors now appropriately state that “ULT in patients with gout was ASSOCIATED with lower risks of hospitalized stroke and all-cause mortality.” However, in the very next sentence they continiue to state that “A subgroup analysis revealed that uricosuric agents HAVE A SIGNIFICANT EFFECT on the decrease of hospitalized stroke, and XO inhibitors HAVE A BENEFICIAL EFFECT on survival.” This causal statement is simply not the case from the data; instead the term “association” should be applied here as it is in the first sentence. The authors need to go through the manuscript and strip any inference of causality out of the interpretation of their data; they may leave such inferences in more speculative sections where they are discussing the possible meaning of their data.

2. The discussion is still too long! It is 9 paragraphs and no reader will have patience for that. Much of the excessive content comes from the heaping up of examples that could be reduced to a summary sentence with multiple references, which would shorten the manuscript significantly. Honestly, there are very few good reasons for a discussion that focuses on the interpretation of a study to go more than 4 or 5 paragraphs.

7. PLOS authors have the option to publish the peer review history of their article (what does this mean?). If published, this will include your full peer review and any attached files.

Reviewer #1: No

---

## [Author Response · Author response to Decision Letter 2]

4 May 2020

Hans-Peter Brunner-La Rocca, M.D.

Academic Editor

PLOS ONE 

May 4, 2020

Dear Dr. Hans-Peter Brunner-La Rocca:

Re: Document reference No. PONE-D-19-31773R2

Please find attached a revised version of our document “Urate-lowering therapy may mitigate the risks of hospitalized stroke and mortality in patients with gout”. We would like to resubmit for publication as an original article in PLOS ONE. 

Your comments and those of the reviewers were highly insightful and enabled us to improve the quality of our document. In the following pages are our responses to each comment from the reviewer as well as your own comments.

Revisions in the text are shown with tracked changes. We hope that our revisions to the document combined with our accompanying responses will be sufficient to render our document suitable for publication in PLOS ONE. 

Yours sincerely,

Chii-Min Hwu

Faculty of Medicine, National Yang-Ming University School of Medicine, and Section of Endocrinology and Metabolism, Department of Medicine, Taipei Veterans General Hospital

Tel.: +886 2 28757516

Fax: +886 2 28751429

E-Mail: chhwu@vghtpe.gov.tw

Address: No. 201, Sec. 2 Shi-Pai Rd., Chung-Cheng Build. 11F Room522, Taipei 112, Taiwan.

Responses to the comments of Editor 

1. As you can see in the comment by the reviewer, there are remaining issues. It is important that the interpretation of data is not speculative, unless clearly stated as such or stated as hypothesis that needs further tested.

Response: We have gone through the whole paper and corrected the interpretation at page 3, 18, 19, 20 and 21 of the tracked manuscript to make it not speculative. 

2. In addition, I agree with the reviewer that the discussion should be shortened. 

Response: We tried to shorten the discussion by using summary sentence with multiple references at page 19, 20 and 21 to make the mention more concise. 

Responses to the comments of Reviewer 1

1. I thank the authors for their responses and resubmission. The manuscript is improved, but

unfortunately, I do not yet feel that the authors have fully responded to this reviewer’s comments. In my opinion, two specific issues remain to be addressed. There continues to persist language that implies causality between ULT and benefit in the interpretation of the data. It is fine to discuss possible causality in sections where mechanisms are being speculated about, but not in sections where the meaning of the data should be interpreted rigorously. Just as one example, at the beginning of

the Discussion the authors now appropriately state that “ULT in patients with gout was ASSOCIATED with lower risks of hospitalized stroke and all-cause mortality.” However, in the very next sentence they continue to state that “A subgroup analysis revealed that uricosuric agents HAVE A SIGNIFICANT EFFECT on the decrease of hospitalized stroke, and XO inhibitors HAVE A BENEFICIAL EFFECT on survival.” This causal statement is simply not the case from the data; instead the term “association” should be applied here as it is in the first sentence. The authors need to go through the manuscript and strip any inference of causality out of the interpretation of their data; they may leave such inferences in more speculative sections where they are discussing the possible meaning of their data. 

Response: Thank you for your encouragement. We have gone through the whole paper and corrected the interpretation at page 3, 18, 19, 20 and 21 of the tracked manuscript to make it not speculative. 

2. The discussion is still too long! It is 9 paragraphs and no reader will have patience for that. Much of the excessive content comes from the heaping up of examples that could be reduced to a summary sentence with multiple references, which would shorten the manuscript significantly. Honestly, there are very few good reasons for a discussion that focuses on the interpretation of a study to go more than 4 or 5 paragraphs. 

Response: We learned much from your recommendations. We have shortened the discussion by using summary sentence with multiple references at page 19, 20 and 21 to make the mention more concise.

---

## [Editor Report · Decision Letter 3]

14 May 2020

PONE-D-19-31773R3

Urate-lowering therapy may mitigate the risks of hospitalized stroke and mortality in patients with gout

PLOS ONE

Dear Dr. Hwu,

Thank you for submitting your manuscript to PLOS ONE. After careful consideration, we feel that it has merit but does not fully meet PLOS ONE’s publication criteria as it currently stands. Therefore, we invite you to submit a revised version of the manuscript that addresses the points raised during the review process.

I have not sent the revisions to the reviewers again as you addressed the points adequately. However, I have to remind you that there is no type editing by the journal and the text needs to be written in correct English. This is, unfortunately, not yet the case. You really need to have the text revised by an **native English **speaking person. In particular, the revised text contains various mistakes. E.g. where you changed to indicate an association instead of a causative effect you always write only "associated", instead of "were associated" (e.g. the abstract: "Subgroup analyses revealed that uricosuric agents significantly associated with lower risk of hospitalized stroke, and XO inhibitors significantly associated with lower risk of all-cause mortality". It should be "... uricosuric agents **were **significantly associated with lower risk..." etc. There are also other examples where you simply have no verb as part of the sentence. You also tend to omit articles (the, a), which is not good English.

I understand that this is difficult for you, but there is no other option as this is the policy of PLOS ONE.

We would appreciate receiving your revised manuscript by Jun 28 2020 11:59PM. To enhance the reproducibility of your results, we recommend that if applicable you deposit your laboratory protocols in protocols.io, where a protocol can be assigned its own identifier (DOI) such that it can be cited independently in the future. For instructions see: http://journals.plos.org/plosone/s/submission-guidelines#loc-laboratory-protocols

We look forward to receiving your revised manuscript.

Kind regards,

Hans-Peter Brunner-La Rocca, M.D.

Academic Editor

PLOS ONE

---

## [Author Response · Author response to Decision Letter 3]

28 May 2020

Hans-Peter Brunner-La Rocca, M.D.

Academic Editor

PLOS ONE 

May 28, 2020

Dear Dr. Hans-Peter Brunner-La Rocca:

Re: Document reference No. PONE-D-19-31773R2

Please find attached a revised version of our document “Urate-lowering therapy may mitigate the risks of hospitalized stroke and mortality in patients with gout”. We would like to resubmit for publication as an original article in PLOS ONE. 

Your recommendations are very important and enable us to improve the quality of our document. In the following pages are our responses to your comments.

Revisions in the text are shown with tracked changes. We hope that our revisions to the document combined with our accompanying responses will be sufficient to render our document suitable for publication in PLOS ONE. 

Yours sincerely,

Chii-Min Hwu

Faculty of Medicine, National Yang-Ming University School of Medicine, and Section of Endocrinology and Metabolism, Department of Medicine, Taipei Veterans General Hospital

Tel.: +886 2 28757516

Fax: +886 2 28751429

E-Mail: chhwu@vghtpe.gov.tw

Address: No. 201, Sec. 2 Shi-Pai Rd., Chung-Cheng Build. 11F Room522, Taipei 112, Taiwan.

Responses to the comments of Editor 

1. I have not sent the revisions to the reviewers again as you addressed the points adequately. However, I have to remind you that there is no type editing by the journal and the text needs to be written in correct English. This is, unfortunately, not yet the case. You really need to have the text revised by a native English speaking person. In particular, the revised text contains various mistakes. E.g. where you changed to indicate an association instead of a causative effect you always write only "associated", instead of "were associated" (e.g. the abstract: "Subgroup analyses revealed that uricosuric agents significantly associated with lower risk of hospitalized stroke, and XO inhibitors significantly associated with lower risk of all-cause mortality". It should be "... uricosuric agents were significantly associated with lower risk..." etc. There are also other examples where you simply have no verb as part of the sentence. You also tend to omit articles (the, a), which is not good English. I understand that this is difficult for you, but there is no other option as this is the policy of PLOS ONE.

Response: Thank you for your encouragement and recommendations! We have asked native English speaking persons (Katie Fonseca and Sam Nagel) to revise our text, especially focus on some mistakes you raised.

---

## [Editor Report · Decision Letter 4]

5 Jun 2020

Urate-lowering therapy may mitigate the risks of hospitalized stroke and mortality in patients with gout

PONE-D-19-31773R4

Dear Dr. Hwu,

We’re pleased to inform you that your manuscript has been judged scientifically suitable for publication and will be formally accepted for publication once it meets all outstanding technical requirements.

Kind regards,

Hans-Peter Brunner-La Rocca, M.D.

Academic Editor

PLOS ONE
---

## [Editor Report · Acceptance letter]

9 Jun 2020

PONE-D-19-31773R4 

Urate-lowering therapy may mitigate the risks of hospitalized stroke and mortality in patients with gout 

Dear Dr. Hwu:

I'm pleased to inform you that your manuscript has been deemed suitable for publication in PLOS ONE. Congratulations! Your manuscript is now with our production department. 

Kind regards, 

on behalf of

Dr. Hans-Peter Brunner-La Rocca 

Academic Editor

PLOS ONE